# Microglia Remodelling and Neuroinflammation Parallel Neuronal Hyperactivation Following Acute Organophosphate Poisoning

**DOI:** 10.3390/ijms23158240

**Published:** 2022-07-26

**Authors:** Julie Somkhit, Constantin Yanicostas, Nadia Soussi-Yanicostas

**Affiliations:** NeuroDiderot, Inserm, Université Paris Cité, F-75019 Paris, France; jullie.somkhit@inserm.fr (J.S.); constantin.yanicostas@inserm.fr (C.Y.)

**Keywords:** microglia, organophosphate poisoning, microglia in vivo imaging, diisopropylfluorophosphate (DFP), zebrafish, cytokines, inflammation

## Abstract

Organophosphate (OP) compounds include highly toxic chemicals widely used both as pesticides and as warfare nerve agents. Existing countermeasures are lifesaving, but do not alleviate all long-term neurological sequelae, making OP poisoning a public health concern worldwide and the search for fully efficient antidotes an urgent need. OPs cause irreversible acetylcholinesterase (AChE) inhibition, inducing the so-called cholinergic syndrome characterized by peripheral manifestations and seizures associated with permanent psychomotor deficits. Besides immediate neurotoxicity, recent data have also identified neuroinflammation and microglia activation as two processes that likely play an important, albeit poorly understood, role in the physiopathology of OP intoxication and its long-term consequences. To gain insight into the response of microglia to OP poisoning, we used a previously described model of diisopropylfluorophosphate (DFP) intoxication of zebrafish larvae. This model reproduces almost all the defects seen in poisoned humans and preclinical models, including AChE inhibition, neuronal epileptiform hyperexcitation, and increased neuronal death. Here, we investigated in vivo the consequences of acute DFP exposure on microglia morphology and behaviour, and on the expression of a set of pro- and anti-inflammatory cytokines. We also used a genetic method of microglial ablation to evaluate the role in the OP-induced neuropathology. We first showed that DFP intoxication rapidly induced deep microglial phenotypic remodelling resembling that seen in M1-type activated macrophages and characterized by an amoeboid morphology, reduced branching, and increased mobility. DFP intoxication also caused massive expression of genes encoding pro-inflammatory cytokines *Il1β*, *Tnfα*, *Il8*, and to a lesser extent, immuno-modulatory cytokine *Il4*, suggesting complex microglial reprogramming that included neuroinflammatory activities. Finally, microglia-depleted larvae were instrumental in showing that microglia were major actors in DFP-induced neuroinflammation and, more importantly, that OP-induced neuronal hyperactivation was markedly reduced in larvae fully devoid of microglia. DFP poisoning rapidly triggered massive microglia-mediated neuroinflammation, probably as a result of DFP-induced neuronal hyperexcitation, which in turn further exacerbated neuronal activation. Microglia are thus a relevant therapeutic target, and identifying substances reducing microglial activation could add efficacy to existing OP antidote cocktails.

## 1. Introduction

Organophosphates (OPs) are a family of organic compounds that includes highly toxic chemicals widely used as pesticides, flame retardants, plasticizers, and to a lesser extent, warfare nerve agents, making OP poisoning a major public health issue worldwide. In recent years, several million intoxications have been reported annually, causing more than 200,000 deaths, primarily suicides [1,2]. OPs bind covalently to acetylcholinesterase (AChE) and irreversibly inhibit its activity, inducing a large accumulation of acetylcholine (ACh) at cholinergic synapses and, therefore, hyperactivation of acetylcholine receptors (AChR), resulting in the so-called cholinergic syndrome. At the neuronal level, AChR overstimulation causes a massive hyperexcitation of cholinergic neurons with large glutamate release and excitotoxicity, and eventually neuronal death [3]. In poisoned humans and preclinical models, acute OP exposure induces peripheral manifestations and impacts the central nervous system, with seizures that may worsen into *status epilepticus* (SE), a life threat if not quickly treated [4]. Besides direct neurotoxicity, OP poisoning can trigger long-term neuronal disorders, such as OP-induced delayed neuropathy (OPIDN), a complex syndrome associating cognitive and psychomotor deficits [5]. Existing countermeasures combine the AChR antagonist atropine with an AChE reactivator oxime such as pralidoxime (2-PAM), and a γ-aminobutyric acid (GABA) receptor agonist of the benzodiazepine family, such as diazepam. However, while such antidote cocktails do mitigate the acute toxicity of OPs, they must be delivered in the very first minutes after exposure and they do not alleviate all long-term neurological deficits [6]. More potent countermeasures with extended therapeutic and temporal windows are thus needed.

Besides neuronal hyperexcitation, data from preclinical models, mostly rodents, have shown that exposure to OPs rapidly induces massive, sustained brain inflammation [7,8,9], a harmful condition that may be at least partly responsible for the long-term psychomotor comorbidities observed in intoxicated patients and animal models [10]. Specifically, it has been clearly established that neuroinflammation creates an environment that is not conducive to the maintenance of brain homeostasis and can promote epileptogenesis [11]. In line with these findings, Gonzales et al. [12] recently showed that microglial cells, the brain-resident macrophages which are key actors in brain inflammation, likely play an important role in the cognitive deficits observed 1 month post-acute OP intoxication in juvenile rats. Importantly, at the therapeutic level, the evidence suggests that agents able to mitigate the OP-induced neuroinflammation could be a promising additional treatment in future antidote cocktails to relieve both the immediate effects of OPs and their long-term consequences.

We previously described a zebrafish model of acute OP intoxication using diisopropylfluorophosphate (DFP), a prototypic OP structurally similar to the G-class nerve agent sarin and widely used in toxicological research due to its moderate toxicity and low volatility [10,13]. This zebrafish model of acute DFP poisoning reproduces almost all the major neuropathological defects seen in exposed humans and preclinical models, including AChE inhibition, massive neuronal excitation leading to epileptiform activity, an imbalance of glutamatergic/GABAergic synaptic activity, and increased neuronal death [14].

Here, we used in vivo imaging and transgenic lines encoding fluorescent reporter proteins, combined with genetic microglia ablation method, to: (1) assess whether our zebrafish model of acute DFP poisoning faithfully reproduced the brain inflammatory response observed in rodent models, (2) study the role played by microglia in DFP-induced neuroinflammation, and (3) investigate the consequences of microglia-mediated inflammation on the subsequent functioning of neuronal networks in individuals acutely exposed to DFP. Our findings confirm that microglia are key players in DFP-induced neuroinflammation and, more importantly, that this inflammatory environment of the brain may further exacerbate DFP-induced excitability of neuronal networks. Our results therefore suggest that microglia are a novel therapeutic target to identify compounds mitigating OP-induced neuroinflammation, and that they could be used to improve the efficacy of existing antidote cocktails.

## 2. Results

### 2.1. DFP Exposure Induced Dramatic Phenotypic Remodelling of Microglia

To investigate the consequences of acute DFP poisoning on the physiology of microglia, we first made use of our well-established zebrafish model of DFP-intoxication [14] and the zebrafish transgenic line Tg[mpeg1:mCherryF], which enables live imaging of these cells in embryos from 3 days post-fertilization (dpf) onward [15] (Figure 1A). In close agreement with previous data [16,17], in 5 dpf larvae almost all microglial cells showed a highly branched morphology, characteristic of the so-called ‘resting’ state, with cells showing several long branches (Figure 1B–B″). In contrast, in age-matched individuals exposed for 6 h to 15 µM DFP, microglial cells displayed a much more rounded morphology and a marked decrease in both the number and length of their branches (Figure 1C–C″).

Precise measurements of several morphological parameters using the Imaris software (Bitplane) confirmed that DFP poisoning caused deep morphological changes of microglial cells. Their mean sphericity (Sp) (see Materials and Methods) was significantly increased (0.79 ± 0.01 vs. 0.63 ± 0.01, *p* < 0.0001) (Figure 1D), and their average surface area (S) was decreased in DFP-treated larvae compared to that observed in controls (840 ± 20 µm^2^ vs. 1118 ± 27 µm^2^, *p* < 0.0001) (Figure 1E). In contrast, the average volume (V) of the cells was roughly similar in DFP-treated and control individuals (1654 µm^3^ ± 56 vs. 1724 µm^3^ ± 58, *p* = 0.64) (Figure 1F). In DFP-exposed larvae, microglia also displayed a reduced number of branches (NB) (0.6 ± 0.03 vs. 4.5 ± 0.1, *p* < 0.0001) (Figure 1G), a decreased mean branch length (ML) (17 µm ± 0.1 vs. 21 µm ± 0.1, *p* < 0.001) (Figure 1H), and thus a decreased mean total branch length (TL) (12.1 µm ± 1.3 vs. 80.0 µm ± 2.1, *p* < 0.0001), compared to those in controls (Figure 1I). A 3D Sholl analysis, a method used to quantify both the extent and complexity of cell branches (Imaris software, Bitplane), fully confirmed that DFP exposure induced major microglial phenotypic changes, with cells showing a decrease in both branch number and length (Figure 1J).

To further characterize the consequences of acute DFP poisoning on microglial morphological changes, we performed a cluster analysis of these cells in control and DFP-treated larvae, based on the five morphological parameters that significantly changed following DFP exposure, namely Sp, S, NB, ML, and TL, as indicated above (see Materials and Methods). Results showed that microglial cells could be clustered into three distinct populations in controls (Figure 1K). The largest cluster comprised 47.1% of the cells and included microglia showing highly branched morphology and low sphericity, likely corresponding to ‘resting’ microglia. The smallest cluster, corresponding to 14.7% of the cells, represented microglia with a low process number and a high sphericity, likely corresponding to M1-type ‘activated’ microglia. The third cluster, which contained 38.2% of the cells, comprised microglia displaying both an intermediate branch number and an intermediate sphericity. We refer to these cells as ‘intermediate’ microglia. In contrast, only two main clusters were observed in DFP-treated larvae (Figure 1K). The larger one, which contained 73.1% of the cells, included microglia resembling ‘activated’ microglia. The other cluster, comprising 25% of the cells, contained microglia showing the ‘intermediate’ phenotype as defined above. It is of note that fewer than 2% of the microglia showed a ‘resting’ phenotype in DFP-exposed larvae, compared to 47% in controls, suggesting that DFP caused a massive, brain-wide microglial activation.

Because the above data indicate that microglia displayed deep morphological changes after 6 h of acute DFP exposure, we next investigated the dynamics of these changes using real-time confocal imaging on live 5 dpf Tg[mpeg1:mCherryF] larvae during 6 h of exposure to 15 µM DFP. In agreement with our previous data [16], prior to DFP addition, microglia were highly branched with several long processes that permanently scan their environment and neighbour cells, and during the first 2 h of DFP exposure, no significant morphological changes could be detected (Figure 2A,B). In contrast, from 2.5 h of exposure, clear remodelling of the cells was observed (Figure 2C,D), which included an increased sphericity and a decrease in the length of the processes. After 3.5 h of exposure, almost all the cells displayed a phenotype resembling ‘activated’ microglia (Figure 2E,F). Excerpts of these representative videos are shown in Appendix A (https://urlz.fr/iNwl, accessed on 11 July 2022) for 5 dpf wild-type embryos and Appendix A for 5 dpf DFP-treated embryos (https://urlz.fr/iNz4, accessed on 11 July 2022). We next tracked microglia in WT and DFP-treated embryos and measured the distance travelled by these cells (Appendix A). Results show microglia travelled a greater distance in DFP-treated embryos than in WT larvae (Appendix A).

### 2.2. DFP Exposure Induced Microglia-Mediated Overexpression of Inflammatory Cytokines

Microglia phenotypic changes observed in DFP-exposed larvae were highly reminiscent of those of ‘activated’ M1-type microglia observed not only in human epileptic brains [18], but also in DFP-exposed rats [19].

To confirm that the remodelling of microglia observed in DFP-treated larvae does reflect an M1-like inflammatory type of activation of these cells, we next investigated, by qRT-PCR analysis of whole-body RNAs, the expression levels of transcripts encoding a set of pro-inflammatory (*Il1β* and *Il8*) and immuno-modulatory cytokines (*Il4*), before and at different time points during a 6 h exposure to either 1% DMSO or 15 µM DFP. In close agreement with the results obtained in preclinical models of DFP intoxication [20,21,22], our data first revealed a massive expression of *il1β* (fold change (fc): 413 ± 67, *p* < 0.0001) and *il8* (fc: 44 ± 6, *p* < 0.0005), together with a significantly increased expression of *il4* transcripts (fc: 2.9 ± 0.4, *p* < 0.001) after 6 h of DFP exposure (Figure 3A). Moreover, precise timing of the expression of these RNAs during DFP exposure indicated that significantly increased expression of *il8* (fc: 1.6 ± 0.2, *p* < 0.05) and *il1β* (fc: 10.4 ± 2.4, *p* < 0.01) transcripts was observed after 1 h and 2 h of exposure, respectively, prior to increasing gradually after 3 h (fc: 3.3 ± 0.7, *p* < 0.001 and fc: 18.5 ± 3, *p* < 0.001), 4 h (fc: 26 ± 9, *p* < 0.001 and fc: 217 ± 28, *p* < 0.001), and 5 h of exposure (fc: 82 ± 9.7, *p* < 0.001 and fc: 125 ± 15.79, *p* < 0.001) (Figure 3). In contrast, increased expression of *il4* transcripts was delayed compared to that of *il1β* and *il8,* starting after 5 h of exposure to DFP (fc: 5.3 ± 0.56, *p* < 0.001) (Figure 3).

To verify that the massive overexpression of RNAs encoding pro-inflammatory cytokines observed in DFP-exposed larvae did reflect brain inflammation, we next investigated, by qRT-PCR analysis of RNAs extracted from dissected brains, the expression levels of transcripts encoding the same set of cytokines (*Il1β*, *II8*, and *Il4*) in larvae exposed for 6 h to either DMSO or DFP. Results confirmed that a 6 h DFP exposure induced massive expression of *il1β* (fc: 189 ± 37, *p* < 0.01) and *il8* (fc: 42 ± 19, *p* < 0.05), and increased expression of *il4* transcripts (fc: 2.9 ± 0.7, *p* < 0.05) in the brain of exposed larvae (Figure 4), confirming that DFP exposure induced bona fide brain inflammation.

### 2.3. Kinetics of Neuronal Activity in DFP-Exposed Larvae

We previously showed that larvae exposed to 15 µM DFP displayed massive neuronal hyperactivation from 1 h to 1.5 h of exposure to DFP [14]. To further investigate this point, we studied, by qRT-PCR analysis of RNAs extracted from larvae at different time points of DFP exposure, the temporal expression profile of *fosab*, an immediate early gene (IEG) whose expression is an early, sensitive marker of neuronal activation, especially epileptiform seizures [23]. A significantly increased expression of *fosab* transcripts was detected in larvae exposed to DFP for 1 h, (fc: 2.89 ± 0.4, *p* < 0.01), which then gradually increased over the next 5 h (fc: 21.7 ± 2.9, *p* < 0.001) (Figure 5). This result suggests, in agreement with our previous calcium imaging data and the results in rodent models [8,14,21] that neuronal hyperactivation in zebrafish larvae is an early consequence of DFP poisoning that has already started as early as 1 h post-exposure.

### 2.4. Inflammatory Cytokines Expression in DFP-Treated Larvae without Microglia

Two glial cell types mediate brain inflammation, including that induced by acute DFP exposure: microglial cells and astrocytes [24,25]. Accordingly, we next undertook to assess the role played by microglia in the neuroinflammatory process induced by DFP exposure. To this end, we analysed the expression levels of the same three cytokine RNAs and *tnfα,* which encodes one of the main pro-inflammatory cytokines, in larvae fully devoid of microglia as the result of morpholino-oligonucleotide-mediated inactivation of the *pU.1* gene, hereafter referred to as *pU.1* morphants [26]. Results indicated that *il1β, il8*, and *tnfα* transcripts were still overexpressed in *pU.1* morphants exposed for 6 h to DFP, albeit at markedly lower levels than observed in their wild-type counterparts (fc: 134 ± 44.3 vs. 413 ± 94, *p* < 0.01, fc: 15.7 ± 4.7 vs. 44 ± 6, *p* < 0.001, and fc: 4.8 ± 1.6 vs. 22.7 ± 4.3, *p* < 0.01, respectively), suggesting that while microglial cells are important players in DFP-induced neuroinflammation, other cells are also involved in the process. In contrast, no overexpression of *il4* was detected in *pU.1* morphants exposed to DFP (fc: 1.2 ± 0.2, *p* = 0.81) (Figure 6), supporting the hypothesis that microglia were the main cell type overexpressing this cytokine following DFP exposure.

### 2.5. DFP-Induced Neuronal Hyperactivation Was Markedly Reduced in Larvae without Microglia

It has long been known that brain inflammation creates an environment that favours neuronal hyperexcitation and epileptogenesis [27]. Therefore, we next set out to evaluate the consequences of microglia activation and subsequent inflammation on the neuropathological processes induced by DFP poisoning. For this purpose, we used pU.1 morphants lacking microglia and showing reduced inflammatory response to DFP to study the consequences of the absence of microglia on DFP-induced neuronal activation as revealed by fosab transcript expression.

The results indicated that *fosab* RNAs were still overexpressed in *pU.1* morphants exposed to DFP, albeit at significantly reduced levels when compared to that observed in their wild-type counterparts (fc: 7.3 ± 1.5 vs. 22.2 ± 2.9, *p* < 0.001) (Figure 7), suggesting that DFP-induced neuronal hyperactivation was markedly reduced in larvae without microglia.

## 3. Discussion

The first important finding of the present study is that acute DFP poisoning in zebrafish larvae rapidly triggered the activation of microglial cells and the synthesis of inflammatory mediators. Here we used in vivo imaging of microglia to describe the dynamics of microglia/macrophage activation after DFP exposure. These microglial morphological changes comprised a rounding of the cells and a decrease in both the number and length of their branches, associated with an increased distance travelled by microglial cell bodies. Interestingly, our cluster analysis further revealed the extent of this remodelling with the population of resting microglia, which decreased from 47.1% to 2% after 6 h of exposure, while that of activated microglia increased from 14.7% to 73.1% over the same period. Such phenotypic remodelling of microglia/macrophages has already been described, and is characteristic of cells committed to an M1-like macrophage activation observed in different brain injury situations, including OP poisoning and various forms of epilepsy [18,19]. We previously showed that zebrafish larvae exposed for 6 h to DFP displayed a marked neuronal hyperexcitation, likely due to a shift in the synaptic excitation/inhibition balance towards an excitatory state associated with an increased number of apoptotic neurons in the brain [14]. The deep remodelling of microglia observed in DFP-exposed larvae might thus reflect an increase in the phagocytic capacities of these cells to cope with the increasing neuronal death and synaptic pruning. Moreover, consistent with the deep remodelling of microglia, we also showed that DFP induced a massive expression of inflammatory cytokines *Il1β*, *Tnfα*, and *Il8*, confirming that DFP poisoning induced a massive and brain-wide activation of microglial cells towards an M1-like inflammatory phenotype. These results are in close agreement with the data from rodent models of DFP poisoning, which showed that DFP exposure triggers a robust neuroinflammatory response resulting from the activation of both microglia and astrocytes [10,19,21,28,29,30,31].

In preclinical rodent models, several teams have reported that DFP poisoning triggers a neuroinflammatory response as early as 1–2 h post-exposure [21,29], making this response an early event in the physiopathology of OP poisoning. Using real-time recording of microglial morphological changes in live transgenic Tg[mpeg1:mCherryF] larvae, we first showed here that phenotypic reprogramming could be detected after approximately 2.5 h of DFP exposure and was clearly seen after 3.5 h of exposure. This result was further refined by the qRT-PCR analysis of the expression of RNAs encoding inflammatory cytokine, which showed significant increases of *il8* and *il1β* transcripts after 1 h and 2 h of DFP exposure, respectively. In the zebrafish DFP model, the increased expression of both *il8* and *fosab* RNAs was detected after 1 h of exposure, making it difficult to establish a causal link between neuronal hyperactivation and neuroinflammation. Thus, as was shown in rodents, in the zebrafish acute DFP model, microglia-mediated neuroinflammation is an early event in OP poisoning that starts as early as 1 h post-exposure. Several authors have hypothesized that neuroinflammation of the brain observed after DFP exposure, and more generally OP poisoning, is the consequence of the excitotoxicity induced by the massive release of glutamate after the overstimulation of AChRs in brain neurons. Data have shown that the severity of seizures in rodents acutely exposed to DFP is positively correlated with the neuroinflammatory response [32]. Moreover, administration of either a low dose of anaesthetic urethane or diazepam to rats during early stages of DFP intoxication, 1 h or 10 min, respectively, markedly mitigated not only neuronal hyperactivation, but also microglia activation and astrogliosis [28,30]. More generally, it has long been known that epileptic seizures trigger a marked neuroinflammatory response that includes the differentiation of microglial cells towards an activated state and the production of inflammatory mediators [33,34]. In human patients with pharmaco-resistant epilepsy, post-mortem analysis of brain tissues revealed a significant microglial activation, which was directly correlated with seizure severity [18,34]. Thus, although the hypothesis of a direct effect of DFP on microglia activation cannot be formally ruled out, the massive microglia-mediated inflammation observed in DFP-exposed larvae is likely caused, directly or indirectly, by the overactivation of cholinergic neuronal networks in the brain.

Using a genetic method, we produced larvae completely devoid of microglia and showed that they displayed a markedly reduced inflammatory response to DFP compared to that of their wild-type counterparts. In particular, we observed an approximately four-fold reduction in the expression levels of *il1β*, *tnfα*, and *il8* transcripts, strong evidence that microglia play a major role in the inflammatory response following DFP poisoning. However, the inflammatory response observed in larvae without microglia, albeit reduced, also suggests that other cell types are involved in the process, likely activated astrocytes. Thus, as already described in DFP-exposed rats [19] and mice [35], our data suggest that DFP-induced neuroinflammation in zebrafish larvae is first mainly mediated by the activation of microglia. In contrast, the expression of RNAs encoding the immuno-modulatory *Il4* cytokine, which was increased in the brain of larvae exposed for 5 h and 6 h to DFP, was not augmented in microglia-depleted individuals similarly exposed, suggesting that overexpression of this cytokine is mainly mediated by microglial cells in exposed larvae. This result also suggests that following DFP poisoning, microglia first respond through M1-like activation and synthesis of inflammatory mediators, followed a few hours later by the expression of regulatory mediators, possibly reflecting an attempt by these cells to restore brain homeostasis. Previous results had already shown that in addition to inflammatory substances, production of anti-inflammatory mediators is also increased in microglia after epileptic seizures, demonstrating that the microglial response to seizures is not limited to the classic M1-like pro-inflammatory activation [36]. However, it is not known whether the same or distinct microglial populations are involved in the two types of responses. Future work will need to clarify this point.

Using *pU.1* morphants, which are devoid of microglia and displayed a significantly reduced inflammatory response to DFP intoxication, we showed that DFP-induced overexpression of *fosab* RNAs was significantly reduced in larvae without microglia. This result suggests that while neuronal activation was first a direct consequence of DFP-induced CNS AChR overstimulation, inflammatory mediators synthesized by activated microglia further exacerbated neuronal activation. Moreover, the relationship between epileptic seizures and brain inflammation is complex, since besides the neuronal activation-induced neuroinflammatory process mentioned above, inflammatory stimuli have also been identified as causative agents of epileptogenesis [37,38,39,40], suggesting a possible vicious circle involving the two processes. In particular, it has been shown that inflammatory *Il1β* is widely implicated in epileptogenesis [27,33,35] and, more importantly, that pharmacological inhibition of *Il1β* signalling with anakinra, a modified recombinant isoform of the human *Il1R* agonist, *Il1Ra,* was shown to drastically reduce seizure numbers in epileptic patients unresponsive to conventional anti-epileptic drugs [41,42,43,44]. In particular, data suggested that *Il1β* might play a role in the pathophysiology of epilepsy through increasing glutamatergic signalling [45] and *N*-methyl-d-aspartate (NMDA) receptor activity [46], and decreasing GABAergic transmission [47].

## 4. Materials and Methods

**Animals.** Zebrafish were kept at 26–28 °C in a 14 h light/10 h dark cycle. Larvae were collected by natural spawning and raised in E3 medium at 28.5 °C [48]. To inhibit embryo pigmentation, 0.003% 1-phenyl-2-thiourea (PTU) was added at 1 day post-fertilization (dpf). The Tg[HuC:GCaMP5G] transgenic line [49], a gift from Dr. George Debrégeas (Laboratoire Jean Perrin, Paris), were raised in our facility. All animal experiments were conducted at the French National Institute of Health and Medical Research (Inserm) UMR 1141 in Paris in accordance with European Union guidelines for the handling of laboratory animals (https://ec.europa.eu/environment/chemicals/lab_animals/home_en.htm, accessed on 11 July 2022). They were approved by the Direction Départementale de la Protection des Populations de Paris and the French Animal Ethics Committee under reference No. 2012-15/676-0069.

**Drug treatment**. Diisopropylfluorophosphate (DFP) was purchased from Sigma Aldrich (St. Louis, MO, USA). A stock solution (5.46 mM), made in DMSO and stored at −20 °C, was diluted extemporaneously to 15 μM in 1% DMSO/E3 medium. Control zebrafish larvae were treated with 1% DMSO/E3 medium.

**RNA isolation and quantitative RT-PCR**. For RNA isolation, whole larvae or dissected brains were homogenized using a syringe equipped with a 26G needle (7 larvae or 10 brains per sample) using the RNA XS Plus kit (Qiagen, Hilden, Germany). Following RNA quantification with a Nanodrop 2000 (ThermoScientific, Waltham, MA, USA) and RNA integrity assessment using denaturing gel electrophoresis, total RNAs (1 µg) were reverse-transcribed into cDNAs using the iScript cDNA Synthesis Kit (Bio-Rad, Kabelsketal, Germany) and qPCRs were performed using iQ SYBR Green Supermix (Bio-Rad, Kabelsketal, Germany). Samples were run in triplicate and expression levels of the studied genes were normalised to that of the *tbp* gene. The primers (Eurofins Genomics, Ebersberg, Germany) used are listed in Appendix A.

**Live imaging of microglial cells**. The transgenic line Tg[mpeg1:mCherryF] was used for all confocal live imaging experiments. In total, 5 dpf larvae were mounted in 1.2% low melting-point agarose containing 300 µM pancuronium bromide (PB) (Sigma) to paralyse the larvae, and either 1% DMSO or 15 µM DFP. They were then imaged for 6 h using a Leica SP8 confocal scanning laser microscope equipped with either a Leica 20×/0.75 multi-immersion objective or an Olympus 40×/1.1 water objective.

**Analysis and quantification of microglial morphological parameters.** For all microglia analysed (327 cells in control and 294 in DFP-treated larvae), six morphological parameters were determined using the Imaris software (Bitplane): sphericity (Sp), calculated as the ratio of the measured surface area of a cell (*A*_c_) to the surface area of a sphere of the same volume (*V*_c_), Sp = (π13×(6Vc)23)/Ac, ranging from 0 (fully disorganised shape) to 1 (perfect sphere); surface area (S); volume (V); mean number of branches (NB); mean branch length (ML); and total branch length (TL).

**Clustering of microglial cells according to morphological criteria.** Among the six morphological parameters defined above, we selected five that changed significantly after 6 h of exposure to 15 µM DFP: Sp, S, NB, ML, and TL. We then defined ranges for each one to characterise the microglial populations. For branched ‘resting’ microglial cells, the ranges were: Sp ≤ 0.6, S ≥ 2400 µm^2^, NB ≥ 6, ML ≥ 19 µm, and TL ≥ 90 µm. For amoeboid ‘activated’ microglial cells, the ranges were: Sp ≥ 0.8, S ≤ 1500 µm^2^, NB ≤ 2, ML ≤ 7 µm, and TL ≤ 20 µm. All microglia in at least three of these five ranges were identified as ‘activated’ or ‘resting’; the remaining cells were classified as ‘intermediate’ microglia.

**Statistics**. Statistical analyses were performed using GraphPad Prism 8.4.3.686 (https://www.graphpad.com/scientific-software/prism/, accessed on 11 July 2022). Data were first challenged for normality using the Shapiro–Wilk test. Data with a normal distribution were analysed with a two-tailed unpaired *t*-test with or without Welch’s correction, depending on the variance difference of each sample. For the statistical analysis of the results obtained with the *pU.1* morphants, treated or not, with DFP, Anova tests were used.

Data not showing a normal distribution were analysed using a two-tailed Mann–Whitney test. All graphs show mean ± SEM.

## 5. Conclusions

In conclusion, our results confirm that the zebrafish model of acute DFP poisoning precisely reproduces the key pathological features observed in rodent preclinical models, including AChE inhibition, epileptiform seizures, neuronal death, and microglia-mediated brain inflammation. In addition, we found that larvae lacking microglia displayed markedly reduced neuronal activation following DFP exposure, suggesting that microglia-mediated neuroinflammation further potentiates DFP-induced neuronal network hyperactivation. Microglia therefore appear as a key part of a vicious circle involving neuronal activation and neuroinflammation following DFP poisoning. These cells could thus be a therapeutic target to identify substances mitigating neuroinflammatory processes. They could thereby add to existing antidote cocktails and improve their efficacy.

## Figures and Tables

**Figure 1 ijms-23-08240-f001:**
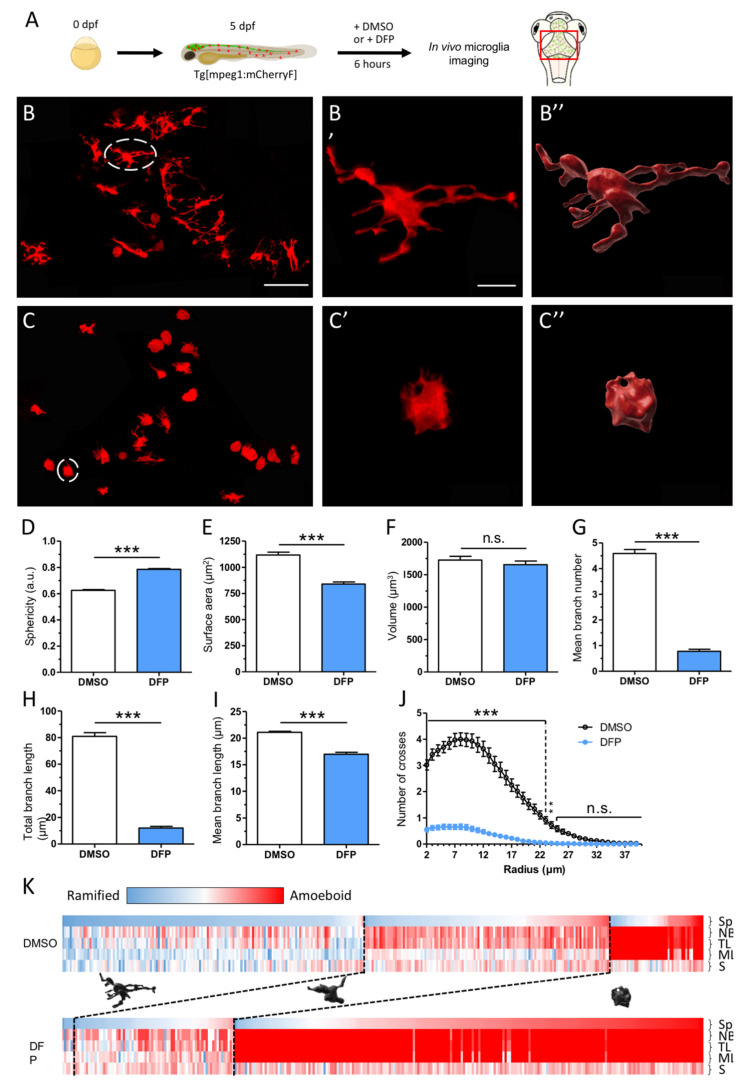
DFP exposure induced major morphological changes of microglial cells. (**A**) Scheme of the experimental set up. Live 5 dpf Tg[mpeg1:mCherryF] larvae were exposed to either 1% DMSO (control) or 15 µM DFP for 6 h and then imaged with a confocal microscope. The region of interest is framed in red. (**B**–**C**) Representative microglial cells in control (**B**) and DFP-treated larvae. (**C**) Scale bar: 50 µm. (**B′**,**B″**) Magnification of the white-circled microglia from a control larva (**B′**), and corresponding 3D reconstruction (**B″**). (**C′**,**C″**) Magnification of the white-circled microglia from a DFP-treated larva (**C′**), and corresponding 3D reconstruction (**C″**). Scale bar: 10 µm. (**D**–**I**) Changes in microglia morphological parameters: sphericity (Sp) (scaled from 0, fully disordered morphology, to 1, perfect sphericity) (**D**), surface area (S) (**E**), volume (V) (**F**), mean branch number (NB) (**G**), total branch length (TL) (**H**), and mean branch length (ML) (**I**) in control (DMSO) (*N* = 13 embryos, *n* = 327 cells) and DFP-treated larvae (DFP) (*N* = 14 larvae, *n* = 294 cells). (**J**) Sholl analysis of microglia branch complexity in control (black) and DFP-exposed larvae (blue). Error bars on all graphs represent the standard error of the mean (SEM). Statistics: ***, *p* < 0.001; n.s., not significant. (**K**) Clustering of microglial cell populations in control (DMSO) and DFP-exposed larvae (DFP), using five of the previously described morphological parameters (Sp, NB, TL, ML, and S) (see Materials and Methods). Each column corresponds to a single microglial cell, and each parameter is scaled from black (‘resting’ state) to red (‘activated’ state); black dotted lines separate the different microglial populations (ramified, transitional, and amoeboid).

**Figure 2 ijms-23-08240-f002:**
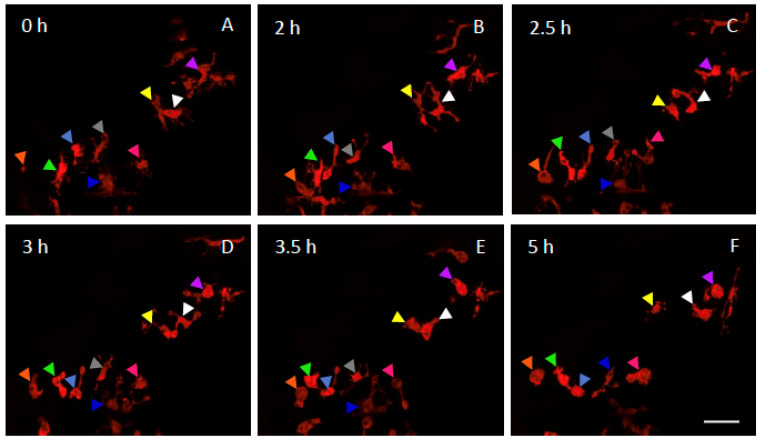
Dynamics of microglia remodelling in DFP-exposed embryos. Selected snapshot views of live imaging of microglia showing the dynamics of microglia morphological changes in a living 5 dpf Tg[mpeg1:mCherryF] larva, before (0 h) and at different time points of a 6 h exposure to 15 µM DFP. Coloured arrowheads indicate individual microglial cells. Scale bar represents 20 µm.

**Figure 3 ijms-23-08240-f003:**
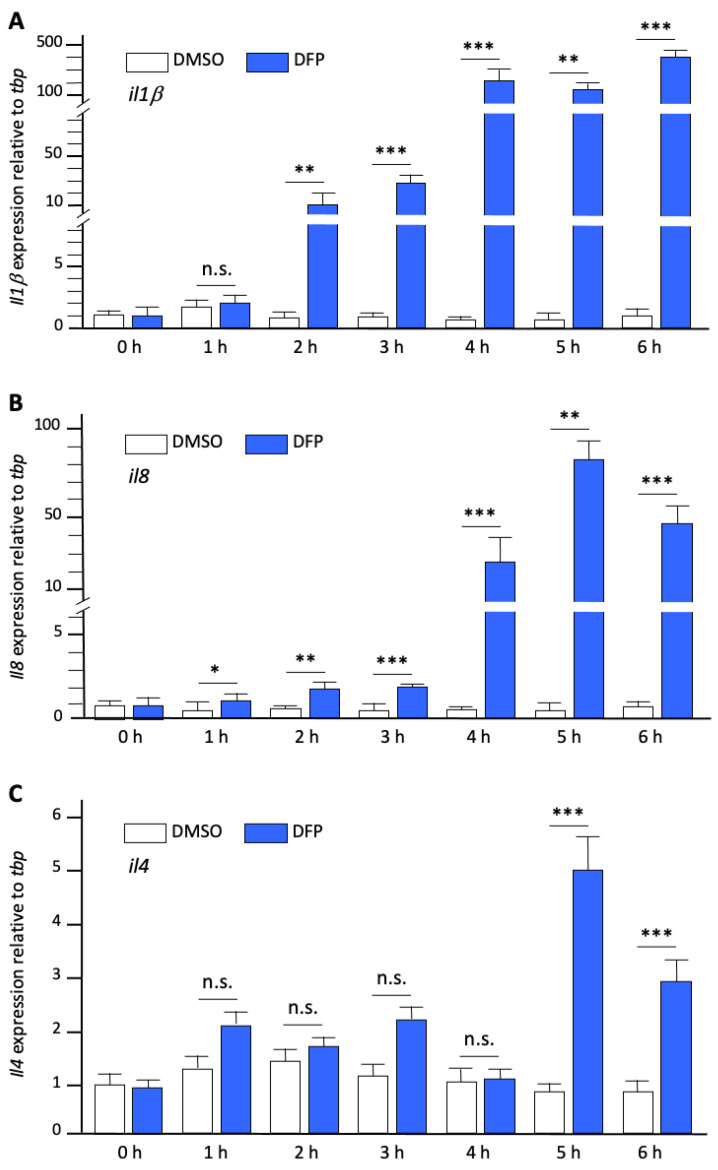
DFP exposure induced massive expression of pro-inflammatory cytokines (**A**–**C**). Expression levels of transcripts encoding cytokines *Il1β* (**A**), *Il8* (**B**), and *Il4* (**C**) relative to that of reference *tbp* transcripts, in RNAs from control (DMSO) and DFP-exposed larvae (DFP) at different exposure times. In each condition, *N* = 8 samples, *n* = 7 larvae/sample. Error bars on all graphs represent the standard error of the mean (SEM). Statistics: *, *p* < 0.05; **, *p* < 0.01; ***, *p* < 0.001; n.s., not significant.

**Figure 4 ijms-23-08240-f004:**
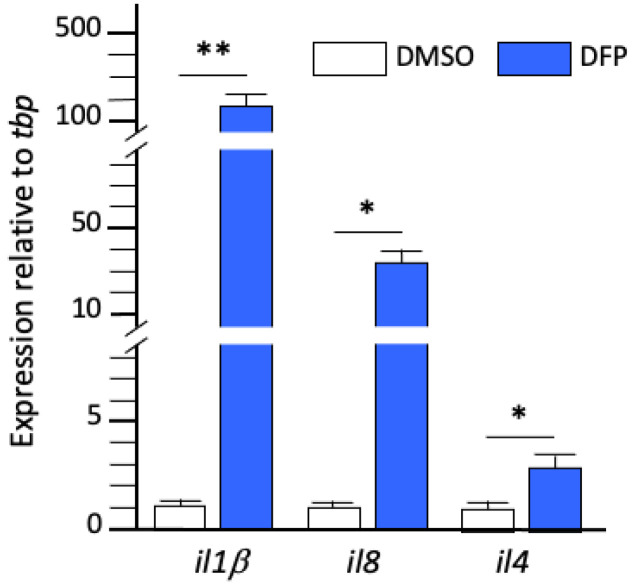
DFP exposure induced massive brain inflammation. Expression levels of transcripts encoding cytokines *Il1β*, *Il8*, and *Il4*, relative to that of reference *tbp* transcripts, in dissected brain RNAs from larvae exposed for 6 h to 15 µM DFP or DMSO (in each condition, *N* = 5 samples, *n* = 10 brains/sample). Error bars on all graphs represent the standard error of the mean (SEM). Statistics: *, *p* < 0.05; **, *p* < 0.01; n.s., not significant.

**Figure 5 ijms-23-08240-f005:**
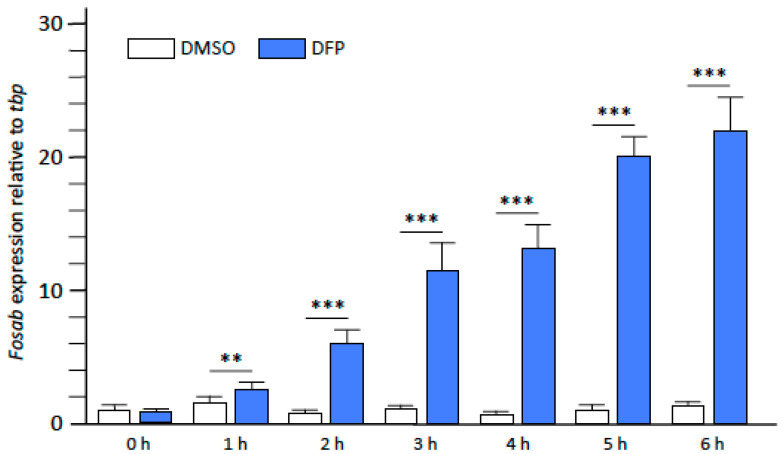
DFP exposure rapidly induced massive neuronal activation. Expression levels of *fosab* RNA relative to that of *tbp* RNA transcripts, from control (DMSO) and DFP-exposed (DFP) larvae, at different time points of exposure (in each condition, *N* = 8 samples, *n* = 7 larvae/sample). Error bars on all graphs represent the standard error of the mean (SEM). Statistics: **, *p* < 0.01; ***, *p* < 0.001.

**Figure 6 ijms-23-08240-f006:**
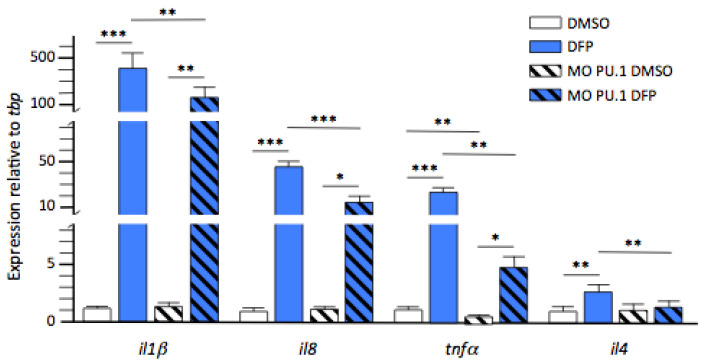
Microglia are key players in DFP-induced inflammation. Expression levels of transcripts encoding cytokines *Il1β*, *Il8*, *Il4,* and *Tnfα* relative to that of reference *tbp* transcripts, from control larvae and *pU.1* morphants exposed for 6 h to either 1% DMSO (DMSO) or 15 µM DFP (DFP) (in each condition, *N* = 7–8 samples, *n* = 7 larvae/sample). Error bars on all graphs represent the standard error of the mean (SEM). Only statistically significant differences between samples are shown: *, *p* < 0.05; **, *p* < 0.01; ***, *p* < 0.001.

**Figure 7 ijms-23-08240-f007:**
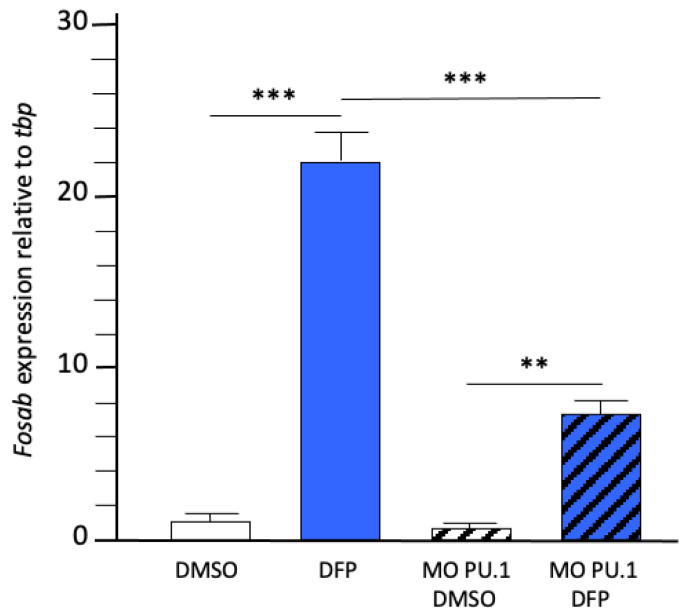
Microglial depletion markedly reduced DFP-induced neuronal hyperactivation. Expression levels of *fosab* RNAs relative to that of reference *tbp* transcripts, from control larvae and *pU.1* morphants exposed for 6 h to either 1% DMSO (DMSO) or 15 µM DFP (DFP) (in each condition, *N* = 7–8 samples, *n* = 7 larvae/sample). Error bars on all graphs represent the standard error of the mean (SEM). Statistics: **, *p* < 0.01; ***, *p* < 0.001.

## Data Availability

The data presented in this study are available on request from the corresponding author.

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
