# Peer review of "Microglia Remodelling and Neuroinflammation Parallel Neuronal Hyperactivation Following Acute Organophosphate Poisoning"

_ijms, 2022, doi:10.3390/ijms23158240_

Round 1
Reviewer 1 Report
The manuscript by Somkhit and colleagues describe the role of microglia in acute organophosphate poisoning. Authors tried to reveal the mechanisms underlying the microglia remodeling and neuroinflammation parallel neuronal hyperactivation in a model of DFP intoxication of zebrafish larvae. However, the link is weak, and more evidence are needed for making any solid conclusions.
It is not surprise that microglia play an important role in DFP intoxication. Results of IL1B, IL8, and IL4 are not enough to persuade the audience, especially the gene expression levels are still very high in the group of MO PU.1 DFP in Fig.5. In addition, protein levels and gene expression levels sometimes do not match. It is also difficult to rule out the effects from other cell types.
If possible, authors will need to check other microglial genes that related to diseases or morphology. Please check this paper: PMID:31508850. Author might consider performing RNA sequencing for a better conclusion of the microglial signatures/profiles.
We need to know how many and what are the differentially expressed genes that related to this specific model and what we can learn from the results. Is the microglial signature in this DFP intoxication model similar to others neurodegenerative diseases (PMID: 28930663, 28602351)?
Author Response
Reply to the comments of Reviewer#1
We thank this reviewer for the careful reading of our manuscript and insightful comments and suggestions, which have been very useful and allow improvement. In the new version of our article, we have taken account of part of the comments and/or suggestions made by this reviewer and have sought to fill the gaps and weaknesses pointed out, as detailed below.
Replies to major points
- About the high expression levels of inflammatory cytokine genes in MO.pU1 larvae exposed to DFP
It has been shown in numerous studies (PMID: 34622217, PMID: 33529768, PMID: 30905768 ) that two populations of glial cells are key mediators of brain inflammation following acute DFP exposure in rodents, microglia and astrocytes, and this response appears to be conserved in zebrafish, even though astrocytes, stricto sensu, have not been formerly described in this vertebrate. Nonetheless, GFAP-expressing glial cells, i.e. astrocyte-like cells, have been observed in zebrafish, and are probably responsible for the residual inflammation observed in MO pU.1 individuals exposed to DFP. Moreover, even though other glial cells are involved, our data showed that microglia are key players in brain inflammation following DFP exposure, thus confirming for the first time in zebrafish the results obtained in rodents.
We do not understand the point: “protein levels and gene expression levels sometimes do not match“, given that at no time have we compared the expression of the genes encoding the various cytokines studied in this work with that of the encoded proteins. Throughout this article, we only show qRT-PCR results. Nevertheless, we have modified the Results section to make this point clearer.
- About further characterizing the microglial signature in DFP-exposed larvae
Following the suggestion of this reviewer, to further characterize the neuroinflammatory response to DFP, we have analysed by qRT-PCR the accumulation of another pro-inflammatory cytokine, tnfa transcripts in controls and MO pU.1 larvae, exposed or not to DFP. The new results, which have been included in the new Fig. 5, fully confirmed that microglia are key players in brain inflammation, although again, other cells also play a role. We have modified the Results and Discussion sections to introduce and discuss these new data.
- About performing RNA sequencing for better characterizing microglial signatures
Obviously, single cell RNA-seq analysis of the transcripts expressed by microglial cells would be the “gold standard“ to fully understand the neuroinflammatory response of these cells following exposure to DFP, but also to compare this response to that in other disease situations. Unfortunately, the larval zebrafish brain contains a few tens of microglial cells only and it is currently technically very difficult to envisage isolating these cells to perform single cell RNA-seq. In particular, in PMID:31508850, the authors FACS-sorted microglia from hundreds of zebrafish larvae prior to performing RNA-seq analysis of bulk RNAs, preventing the diversity of these cells from appearing. In addition, the specific microglial signatures in different pathological situations have not yet been clearly defined in zebrafish as they have been in rodents. Therefore, although the single-cell RNA-seq analysis of the microglial signature following acute OP poisoning suggested by this reviewer would be invaluable, it is not feasible in the context of the revision of this article because of the time and workload it implies.
- About what can we learn from the results
Although the question “is the microglial signature in this DFP intoxication model similar to other neurodegenerative diseases" is a very important issue in the field, our purpose was not to investigate this point. Our goals were (i) to verify that our zebrafish model of acute DFP poisoning, the first in this species, faithfully reproduced the defects observed in rodent models, (ii) to confirm the key role played by microglial cells in DFP-induced neuroinflammation, and (iii) to study the consequences of microglia-mediated inflammation on the subsequent excitability of neuronal networks in individuals acutely exposed to DFP.
Changes to the manuscript
As suggested by this reviewer, to further characterize the microglial response to acute DFP exposure, we have analysed the expression of the gene encoding TNFa, one of the main pro-inflammatory cytokines massively expressed by microglial cells in various brain inflammation contexts, including acute DFP exposure in rodents. Using qRT-PCR, we studied the accumulation of this mRNA in WT and MO pU.1 individuals exposed or not to DFP, and the data fully confirmed (i) that massive and bona fide neuroinflammation of the brain was rapidly induced in zebrafish larvae acutely exposed to DFP, and (ii) that microglial cells are key players in this process, although other cells also play a role. These new data have now been included in the new Fig. 5 and in the Results and Discussion sections.

Reviewer 2 Report
The manuscript entitled “Microglia remodelling and neuroinflammation parallel neuronal hyperactivation following acute organophosphate poisoning” aims to gain insight into the response of microglia to OP poisoning. The results are significant, but the presentation is a bit confusing in some parts. The fragments of the manuscript look like they are mixed up. Here are my comments:
Pesticides and warfare agents are not the only applications for OP compounds. It should be corrected.
Lines 73 – 90: It looks more like the Discussion. This part of the Introduction should explain the aims of the study, not the results.
The results are well presented.
The Discussion is good.
The conclusions are supported by the results.
Author Response
Reply to the comments of Reviewer#2
We thank this reviewer for the careful reading of our manuscript and insightful comments and suggestions, which have been very useful and allow improvement. In the new version of our article, we have taken account of all the comments and/or suggestions made by this reviewer and have sought to address the issues pointed out, as detailed below.
Replies to major points
- About the many uses of organophosphorus compounds
We agree that there are OPs other than pesticides and warfare agents and we have modified the introduction to make clear that OPs are a large family that besides pesticides and warfare agents, includes other substances showing environmental toxicity, such as flame retardants and plasticizers.
- About lines 73 - 90
We agree with this comment and have modified this part of the introduction to explain the aims of the study more clearly.
- About the presentation of the results
We have modified the title of the fourth paragraph of the result section which was wrong and confusing, as pointed out by this reviewer: “Inflammatory cytokines and neuronal activity in DFP-treated larvae without microglia” has been replaced by “Inflammatory cytokines expression in DFP-treated larvae without microglia“
Changes to the manuscript
As suggested by this reviewer:
1/ We have modified the introduction to make clear that besides pesticides and nerve gazes, OPs include other substances showing environmental toxicity, such as flame retardants and plasticizers.
2/ We have modified the last paragraph of the introduction to make the objectives of our work clearer.
3/ We have modified the title of the fourth paragraph of the result section.

Reviewer 3 Report
In this interesting article the authors confirm that the zebrafish model of acute DFP 370 intoxication accurately reproduces the key pathological features observed in preclinical rodent models, thus microglia appear as a key part of a vicious circle involving Neuronal activation and neuroinflammation after DFP intoxication.
Some minor issues could improve the manuscript:
1- The objective of the study must be specified
2-All abbreviations must be indicated in all figures.
3- Correct Fig1'
Author Response
Reply to the comments of Reviewer#3
We thank this reviewer for the careful reading of our manuscript and insightful comments and suggestions, which have been very useful and allow improvement. In the new version of our article, we have taken account of all the comments and/or suggestions made by this reviewer and have sought to address the issues pointed out, as detailed below.
Replies to major points
- About the objectives of the study
We agree with the comment and have modified the last part of the introduction to explain the aims of the study more clearly.
- Minor points
- All abbreviations have been indicated in all figures.
- 1' has been corrected.
Changes to the manuscript
As suggested by this reviewer, we have modified the last paragraph of the introduction to make the objectives of our work clearer.

Round 2
Reviewer 1 Report
No more comments and questions.